# Design and Transition of an Emergency E-Learning Pathology Course for Medical Students—Evaluation of a Novel Course Concept

Christopher Holzmann-Littig [1,2,*,†] , Nana Jedlicska [1,†], Marjo Wijnen-Meijer [1], Friederike Liesche-Starnecker [3], Karen Schmidt-Bäse [1], Lutz Renders [2], Katja Weimann [1], Björn Konukiewitz [4] and Jürgen Schlegel [5]

1   TUM Medical Education Center, School of Medicine, Technical University of Munich, 81675 Munich, Germany
2   Department of Nephrology, Klinikum rechts der Isar, School of Medicine, Technical University of Munich, 81675 Munich, Germany
3   Pathology, Medical Faculty, University of Augsburg, 81656 Augsburg, Germany
4   Institute of Pathology, University Hospital Schleswig-Holstein (UKSH), Campus Kiel, 24105 Kiel, Germany
5   Institute of Pathology, School of Medicine, Technical University of Munich, 81675 Munich, Germany
*   Correspondence: christopher.holzmann-littig@mri.tum.de; Tel.: +49-89-4140-8495
†   These authors contributed equally to this work.

**Abstract:** Background: Around the world, the emergency brought about by the COVID-19 pandemic forced medical schools to create numerous e-learning supplements to provide instruction during this crisis. The question now is to determine a way in which to capitalize on this momentum of digitization and harness the medical e-learning content created for the future. We have analyzed the transition of a pathology course to an emergency remote education online course and, in the second step, applied a flipped classroom approach including research skills training. Methods: In the summer semester of 2020, the pathology course at the Technical University of Munich was completely converted to an asynchronous online course. Its content was adapted in winter 2021 and incorporated into a flipped classroom concept in which research skills were taught at the same time. Results: Screencasts and lecture recordings were the most popular asynchronous teaching formats. Students reported developing a higher interest in pathology and research through group work. The amount of content was very challenging for some students. Conclusion: Flipped classroom formats are a viable option when using pre-existing content. We recommend checking such content for technical and didactic quality and optimizing it if necessary. Content on research skills can be combined very well with clinical teaching content.

**Keywords:** remote education; pathology; flipped classroom; medical education; e-learning; COVID-19

## 1. Introduction

In many medical schools, the COVID-19 pandemic forced a prompt transition to entirely digital teaching [1]. In addition, before the crisis, e-learning had already played an important role in medical education. Fundamentally, the term e-learning describes learning in which digital media are used in the presentation and distribution of learning materials or in communication between learners and/or teachers [2]. In medical education, e-learning includes classic digital media, flipped classrooms, communication tools (blogs, chats, etc.), electronic assessments, audio- and video-based tutorials, and interactive teaching formats (e.g., educational games based on digital patient cases) [3,4]. However, the development of content-wise and didactically valuable teaching formats is time-consuming and personnel-intensive; thus, these qualities could no longer be realized in this way during the pandemic. Numerous courses based on the principles of Emergency Remote Education have emerged across the globe [1,5]. In addition, the pathology course at the Technical University of Munich Medical School was redesigned under emergency circumstances. Yet, there are

significant differences between emergency remote education and long-term planned online courses [5]. Emergency remote teaching entails a very short preparation time combined with limited conception options. In addition, altered teaching conditions must be dealt with, as often no phases requiring attendance can be offered whatsoever due to local restrictions/lockdowns. This can lead to the lower quality of remote-teaching courses compared to courses created under normal conditions [5]. At the same time, we aimed to design course content so that it would be suitable for long-term use. With respect to the development of courses, we planned for this transition from the beginning. In flipped classroom approaches, learning materials are provided beforehand so that they can be reflected in the studied content during lessons [6]. Since flipped classrooms are well suited to improving student learning [3], we aimed at a flipped classroom design that could be used under pandemic and non-pandemic circumstances.

Another important issue has been the lack of sufficient research skills-focused training in our medical school. There is also evidence in the literature that medical students sometimes lack adequate research skills-focused training in other medical schools [7–9]. However, there is an agreement on the importance of physicians' competence with respect to using and conducting research [8]. Keeping up with the latest research and being able to critically appraise the scientific literature and incorporate scientific knowledge into clinical decision making is considered crucial in order to be able to make grounded diagnoses and evidence-based therapeutic decisions [10]. This is underlined by the Canadian Medical Education Directives for Specialists (CanMEDS), thus highlighting the importance of research as one of a practitioner's core competencies [11]. The importance of research is also underlined in the German Master Plan for Medical Studies 2020 [12]. Nevertheless, recent studies have shown that only a minority of medical students are inclined to undertake research or even a research career [13,14]. As a result, diverse research-related courses are emerging in curricular and extra-curricular research programs within many medical schools [15–23]. To meet the demand for research skills, we integrated the training of research skills in our pathology course. Furthermore, according to Kaufmann and Mann, "Learning is enhanced when it is relevant, particularly to the solution and understanding of real-life problems and practice" [24]. As previously proposed by Ommering et al., and following the findings of Vereijken et al., we aimed to encourage medical students' interest in research and increase their knowledge regarding research by engaging them in small research projects [8,25]. The theoretical framework for the design of the course was constructed in accordance with Self-Determination Theory (SDT). SDT presents a theory of human motivation that assumes humans' innate tendency to grow and outlines three core psychological needs that foster intrinsic motivation: autonomy, competence, and relatedness. Autonomy refers to the human need to feel in control of one's own behavior and goals. Competence indicates the human desire to have skills and be able to succeed. Relatedness refers to the need to experience a sense of belonging [26]. While designing the online pathology course, we aimed to meet the students' needs and strengthen their feelings of autonomy, competence, and relatedness in order to enhance their intrinsic motivation and interest in research. Thus, in addition to achieving knowledge regarding pathology, the students in this course would be able to gain the skills necessary to work independently in a scientific manner (competence). Their autonomy was to be promoted early on through the flipped classroom approach, and relatedness was to be fostered through working in small groups.

Self-directed learning, an "increasingly essential [method] in the development and maintenance of professional competence" implies "acceptance of personal responsibility for learning" and "personal autonomy" and is regarded as a "method of organizing learning and instruction with the tasks of learning primarily in the learner's control" [27]. This model was to be addressed through largely autonomous, yet professionally guided, learning.

Taken together, the new course design was intended to serve three goals: First, a flipped classroom design for the pathology course that could be used under pandemic and non-pandemic circumstances for long-term use was to be produced. Second, medical

students' interest in research and their knowledge with respect to research were to be improved by engaging them in small research projects. Third, while achieving the latter, the students' feelings of autonomy, competence, and relatedness were to be strengthened in order to enhance their intrinsic motivation and interest in research. The purpose of this work was to investigate the students' satisfaction with the new course concept compared to traditional teaching methods as well as the Emergency Remote Education course. In addition, the students' attitudes towards pathology and research after attending the new course were evaluated. Furthermore, we investigated whether the students lacked direct contact with the lecturers in the new course concept. At the same time, we described the transformation of an originally classical pathology course via an Emergency Remote Education course to a new course model. During the COVID-19 pandemic, many new e-learning course concepts have emerged, including pathology [28–31]. Based on the experience we have gained, we provide suggestions regarding how to create new course concepts out of Emergency Remote Education content and which pitfalls should be considered.

## 2. Materials and Methods

### 2.1. Course Development

The original pathology class took place during the first clinical year of medical school, was mandatory, and was completed with a graded examination. The class consisted of lectures and group seminars and was held face-to-face. It covered the topics of general pathology and neuropathology (all other special pathology topics are included in an integrated lecture in the second clinical year of medical school).

The new General Pathology Online class was established in two developmental phases. It replaced the previous, original pathology course; consequently, this original course can no longer be attended.

2.1.1. Phase 1: Emergency Remote Education [5] Concept ("DigiPath"), Summer Semester 2020

With 31 teaching units, the course covered all previously taught aspects of general and special pathology (Supplemental Table S1). The content was presented in an asynchronous (i.e., it could be completed at any time of the day) e-learning module ("DigiPath"), created as a "Moodle® [32]" course. Screencasts (i.e., videos showing a presentation discussed by a speaker), animated educational videos, or lecture recordings were used. In six units, only lecture slides were presented. This entirely asynchronous approach was necessary, as during the first phase of the pandemic many students were employed as auxiliary staff in the clinic and, therefore, had to be able to attend classes more flexibly. In the screencasts and animated educational videos, teaching content was briefly summarized; these videos usually lasted 4–10 min. This content included different tissue structures, the pathophysiology of strokes, or the structure of nerve cells in connection with demyelinating diseases. In the screencasts, a speaker, usually visible, explained a graphic overview of the topic at hand. In the educational videos, for example, the graphics were drawn during the video and simultaneously explained by a speaker. In the lecture recordings, the previous lecture was given by a lecturer as before, but this was now recorded online and uploaded for asynchronous use; for example, this strategy was applied for the lectures on inflammation and immunology. All videos were produced by the lecturers themselves. At the TUM Medical School, all lecturers had the opportunity to receive advice on technical implementation from an e-learning team at the TUM Medical Education Center (Figure 1, Supplemental Table S1).

Knowledge tests included single- or multiple-choice questions, drag-and-drop tasks, open questions, and matching questions, which were built in between the learning sections. This design was intended to improve the students' sense of competence (as part of the Self Determination Theory) in addition to the efficacy of training [26]. Only after answering the tests correctly were the students able to progress to the next unit. There was no limit to the number of attempts. The week's units were sequentially activated at the beginning

of each week and available until the end of the semester. As self-directed learning should include the opportunity to ask questions [24], the students were offered voluntary virtual meetings with the lecturer at the end of each week. For these virtual question-and-answer meetings, Zoom® [33] was used in the version licensed for our university and complying with local, German, and European data protection regulations. In addition, students could also email their questions to the respective lecturers. A question forum was also set up on the Moodle® [32] platform. Due to the pandemic, final oral examination had to be cancelled.

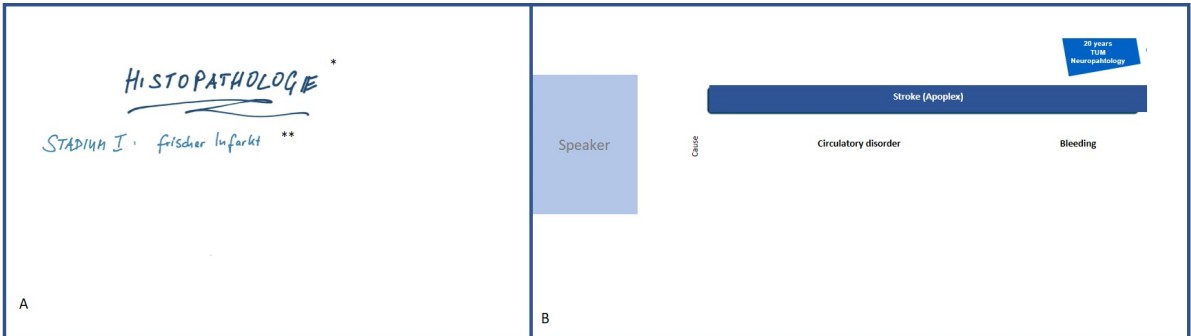

**Figure 1.** Video and screencast example. (**A**) Video on drawing technique; * Histopathology and ** Stage I—acute infarction. (**B**) Scheme of a screencast with speaker and animated content.

2.1.2. Phase 2: Flipped Classroom Concept, Winter Semester 2020/2021

In the next step, a flipped classroom concept was developed, consisting of the asynchronous e-learning ("DigiPath") module followed by seminar module (project phase, "Projects in Pathology") [24] developed by the pathology lecturers. In the project phase, knowledge on specific pathology topics was to be deepened and research skills (scientific thinking, management of scientific projects, and teamwork) were to be trained. The goal was to create long-term, stable teaching programs in pathology at our site that were independent of pandemic waves. Pathology content was planned by the lecturers at the Institute of Pathology. Consultative guidance in planning was provided by the TUM Medical Education Center. For the "Projects in Pathology" phase, the content was developed jointly by the lecturers at the Institute of Pathology, the TUM Center of Key Competencies, and the TUM Medical Education Center. We used Moodle® [32] as the course management system. Kern et al. have proposed a six-step process of curriculum development [34] that we used in the development process. These steps are as follows: the identification of the problem, a (targeted) needs assessment, a definition of goals and objectives, the selection of educational strategies, and the implementation and the evaluation process [34] (Figure 2). The different developmental phases are presented in Figure 3.

Flipped Classroom Concept, Part 1: Asynchronous E-Learning Module "DigiPath"

The asynchronous content was slightly modified, and the units were divided into five subjects: general pathology, inflammation/immunology, general tumorology, special tumorology, and neuropathology. The teaching content was presented exclusively in screencasts and animated educational videos as described above. Due to technical problems with respect to the learning management system, the examination tasks between the units had to be suspended. The course units were activated sequentially on a daily basis to allow their completion by the start of the seminar phase and were available until the end of the semester. A weekly "meet the expert session" was provided, in which cases pertaining to the actual topics were discussed and questions were answered. All content continues to be offered to this day.

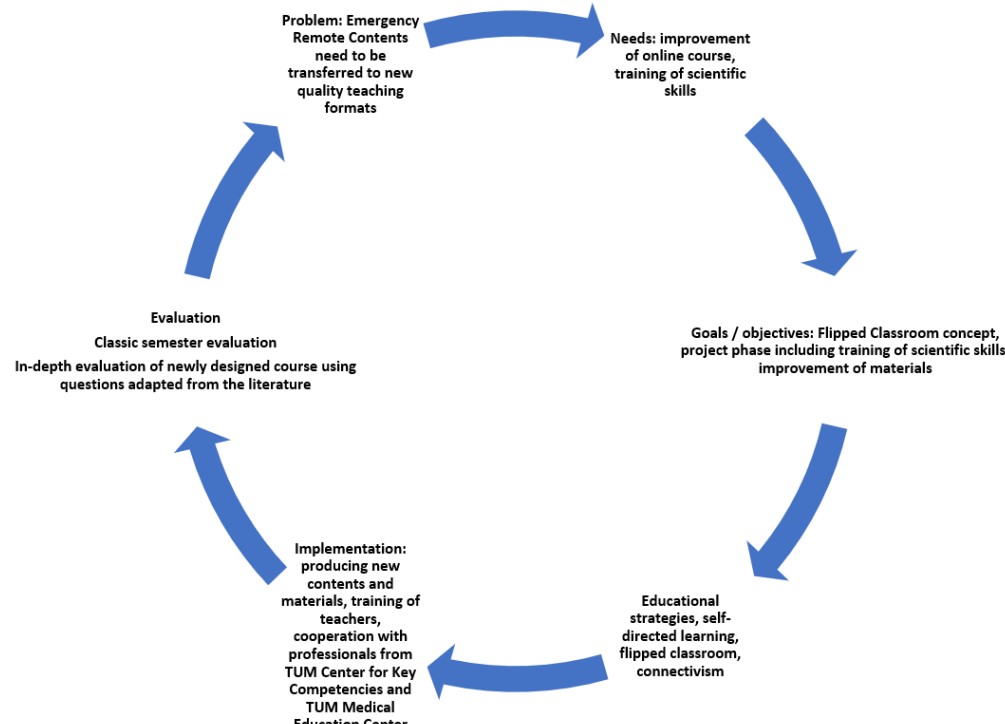

**Figure 2.** Development process in transition phase, adapted from Kern et al. [34].

| Original Pathology Class → | | Emergency Remote Education (5) Concept Summer Semester 2020 → | | Flipped Classroom Concept Winter Semester 2020/2021 | |
|---|---|---|---|---|---|
| Lectures & Seminars | • face-to-face | E-Learning module "DigiPath" | • 31 asynchronous lessons<br>  - 52 screencasts<br>  - 79 e-lecture videos<br>  - 18 lecture notes<br>• Quizzes after each session<br>• Virtual meeting option every Friday | E-Learning module "DigiPath" | • 30 asynchronous lessons<br>  - 38 screencasts<br>• Meet the Expert<br>• No Quizzes |
| | | | | Project phase "Projects in Pathology" | • small group teaching<br>  - 20 Groups with 9 participants each (1 Lecturer and 8 students)<br>• in-depth study of one topic<br>• Training of core competencies:<br>  - scientific thinking<br>  - management of scientific projects<br>  - teamwork<br>• Poster presentation |

**Figure 3.** Transformation of the original class.

Flipped Classroom Concept, Part 2: Project Phase ("Projects in Pathology")

After the planning of the course design, a training program for medical teachers was developed. This program was developed in accordance with the concept that teachers should act as coaches who support the students in their learning process and support the development of the students' research skills. The teachers were trained in the new course materials and in how to conduct group sessions when developing a project. These training programs were organized together with and guided by TUM Center of Key Competencies; materials regarding the teaching of research skills were produced by TUM Medical Education Center. These training sessions were held as face-to-face meetings as part of the regular pathology staff meetings. All training content was also made available to lecturers on the Moodle® [32] e-learning platform.

Ommering et al. (2020) suggested dividing students into small groups and assigning lecturers with research expertise to each group when designing a course with a focus on training research skills [8]. Consequently, 20 groups with a maximum of 8 students each were built. When selecting research teaching content, we followed the graduate attributes of Laidlaw et al. [35], although not all of these could be implemented in a single course. First, each group was assigned one pathology subject that had to be studied in depth. Each topic had been studied in the DigiPath module beforehand. An example of a lesson-planning outline (neuroendocrine tumors vs. neuroendocrine carcinomas) drafted during the planning of the project phase has been provided in Supplemental Table S2. On the basis of this topic, the path from general pathology and pathophysiology background to a clinical diagnosis and therapy was developed using a problem-based [36] approach. Consequently, a scientific poster on this topic was to be created. Furthermore, the skills "management of scientific projects", "teamwork", and "scientific thinking" (literature research, critical reading, and designing scientific posters) were trained. For these skills, the trainers gave explanations and short introductions to the students. Subsequently, the students had to implement this by means of homework, presentations, group work, and the creation of a final poster, whereby the progress was controlled by the trainers during the seminar sessions (Supplemental Table S2). Following the AMEE guide on developing medical students' research skills [37], we used a blended learning concept, which was implemented via case-based learning (using patient-centered cases), and developed a course curriculum integrating research skills and clinical knowledge [37]. Screencasts for preparation at home were developed. Considering the implications that Kaufman and Mann [24] derived from social cognitive theory [38], we created a goal that students had to achieve at the end of the course, thus supporting their learning process. Considering the suggestions of Ommering et al. and in terms of autonomy as part of the Self-Determination Theory [26] (SDT), we gave the students freedom to organize themselves within the group and conduct their research projects individually in order to foster their feelings of autonomy and, thus, enhance their feelings of the ownership of their research, which is seen as important to successfully persist in undertaking a single activity [8]. The feeling of relatedness [26] was to be strengthened through group work. Furthermore, we designed the project-based work to address the theory of self-directed learning [24].

Due to the worsening of the pandemic, the project phase had to be held in virtual classrooms. As presenting and discussing new knowledge promotes deep learning and offers an opportunity to train one's ability to critically appraise the scientific work of others [8], a symposium was planned. Since this face-to-face symposium could not take place, the final posters were discussed within the groups online. Each group discussed and evaluated three posters from the other groups based on the prepared criteria for the quality of a scientific poster (Figure 3).

In summary, the "DigiPath" teaching format and the subsequent "Projects in Pathology" teaching format have completely replaced the original course. The seminar parts and question sessions can and will, as far as the pandemic situation allows, take place face-to-face, while the other units still will be offered virtually. This course is intended to continue to allow flexibility in terms of responding to the pandemic situation by allowing

the seminars and Q&A sessions to be held virtually at any time instead of face-to-face. In this way, we hope to achieve the highest possible stability of pathology teaching in our faculty in the future.

### 2.2. Ethics, Informed Consent

The study was approved by the Ethics Committee of the Medical Faculty of the Technical University of Munich under the code 340/20 S-KH. All participants provided written informed consent statements before answering the questionnaires. All participants were of legal age to take part in the study.

### 2.3. Surveys and Evaluation

The surveys were performed using EvaSys (https://evasys.de/, accessed on 21 December 2022). The pathology course was assessed by the students as part of the general semester evaluation (short evaluation). In addition, students were asked to complete a larger survey for deeper understanding. In this survey, each course section was rated with school marks (ranging from 1 = very well to 6 = unsatisfactory) as this system is generally used in the evaluations of classes in our university (in terms of satisfaction with the course units). This item had also been employed in previous semesters as part of a general evaluation. In order to achieve comparability with previous evaluation results, the average grade of all evaluations of the individual course units from the summer semester 2017 to the winter semester 2019/2020 (total semesters) was calculated. All further scale-based questions were designed with five-point Likert scales, as these are the most commonly used type of Likert scale [39]. Questions on usability and structure were designed based on prior survey items developed by Liu, Lewis, and Herrmann [40–42]. In summer semester 2020, both surveys were sent to all 185 course participants; in winter semester 2020/2021, they were sent to all 157 participants. The surveys took place directly after the end of each semester, i.e., in July 2020 and February 2021. All surveys were closed after 10 days. All students were invited via e-mail. No reminders were sent out so as to avoid reducing respondence in other evaluations and surveys at the end of each semester. Participation in the surveys was voluntary. Since all participants had given their written informed consent, no exclusions were necessary. If individual questions were not answered, these were indicated as missing in the analysis. The survey questions and results can be seen in Supplemental Tables S2 and S3. Free-text comments were translated from German to English for presentation in the manuscript.

### 2.4. Statistics

Statistical analysis was performed using JASP, Version 0.14.1 [43]. For descriptive analyses, mean values with standard deviation and percentages were built and presented as means (SD). The overall satisfaction with the course units was correlated with the different types of teaching formats (e.g., screencasts, lecture recordings, etc.) using point biserial correlation analysis with Pearson's r [44]. A *p*-value of 0.05 was considered statistically significant. Free-text comments were evaluated and categorized by two different team members. Anchor examples were chosen separately and taken for each section after discussing them. Since the course structure differed between summer and winter semesters as described above, and as there was less time available to complete the "DigiPath" course in the winter semester, the results for the asynchronous course sections in the summer and winter semesters are presented separately. Since the student cohorts were predefined by the curriculum and could not be changed, a sample size calculation was not performed for this observational study.

## 3. Results

### 3.1. Participants

A total of 29 students (emergency remote [5] phase, summer semester 2020) and 68 students (transition phase, winter semester 20/21) filled in the short evaluation as part

of the general semester evaluation. A total of 23 students filled in the large questionnaire in the emergency phase, while 27 did so in the second phase. A total of 92% of the students were younger than 25 years. A total of 74% were female; 66% of all the invited students were female.

### 3.2. Overall Satisfaction

The mean grade of all course evaluations of the original pathology course (from the summer semester 2017 to the winter semester 2019/2020) had been 1.9 (the best average evaluation was 1.85 in the winter semester 2017/2018, while the worst average evaluation was 1.98 in the summer semester 2017). In the long survey, the asynchronous teaching units in the new emergency remote education [5] phase were given better ratings, with an average grade of 1.51, followed by 1.98 in the flipped classroom concept. The project phase was given an improved average rating of 1.44. (Since this is the usual practice in our faculty, German school grades were used in student surveys to evaluate satisfaction, with 1 being the best and 6 being the worst.) However, in the short survey as part of the general semester evaluation, a decrease from 1.56 to 2.32 was observed with respect to overall satisfaction (Figure 4, Supplemental Table S4). The individual grades of the course days are listed in Supplemental Table S1.

| Item | Original Course summer semester 2017 – winter semester 2019/2020 (students were obliged to evaluate directly after each unit) | Emergency Remote (5) Concept (Asynchronous E-Learning module only, "DigiPath") | Flipped Classroom Concept | |
|---|---|---|---|---|
| Overall grade (part of general semester evaluation) | | 1.56 | 2.32 | |
| Average grade of course units (long survey and results from original course) | 1.90 | 1.51 | 1.71 | |
| | | | Asynchronous E-Learning module ("DigiPath") | Class Module (Project phase („Projects in Pathology")) |
| | | | 1.98 | 1.44 |

**Figure 4.** Mean grades of courses.

### 3.3. Asynchronous E-Learning Module—"DigiPath"

3.3.1. Satisfaction with Different Teaching Materials

In the emergency remote [5] phase, in terms of the evaluation of the course units screencasts/animated educational videos ($r = -0.515$, $p < 0.001$), case vignettes ($r = -0.482$, $p < 0.001$) and scripts ($r = -0.354$, $p = 0.005$) were inversely correlated with higher (worse) school grades for each unit. Lecture recordings were correlated with higher (worse) school

grades (r = 0.252, *p* = 0.048) for each unit. According to the free-text comments, the video recordings still improved learning success due to the additional explanations provided by lecturers. Units in which only lecture slides were offered received the worst ratings. Accordingly, in the free-text comments, the students stated that they understood significantly less when only lecture slides were offered.

> *"Even though I wouldn't say that I didn't understand [the topics], I took significantly less with me when only slides were uploaded".*

Asking students which teaching formats they considered suitable, screencasts (summer 4.55 [0.74]; winter 4.22 [1.09]/five points on a Likert scale (1 = worst, 5 = best)) and animated educational videos (summer 4.63 [0.60]; winter 4.63 [0.84]) were favored. Lecture recordings were regarded as less suitable (summer 3.79 [1.82]; winter 4.13 [1.23]).

Regarding the preferred teaching units, neuropathology units (which included the most screencasts, as shown in Supplemental Table S1) were mentioned in 86% (summer) and 79% (winter) of the responses. In the free-text comments, short and informative screencasts, animated educational videos with a focus on essential content, short handouts, and clear learning objectives were cited. Shorter videos were regarded as more exciting and easier to memorize. Accordingly, the students suggested that all the material should be offered as short screencasts.

> *"Some lecturers presented their material in a screencast much more concise and shorter than it would probably have been in a lecture. That was good because it allowed the essentials to stick better and it was more exciting to listen to".*

In the free-text comments, it was evident that the students appreciated the asynchronous learning format. A total of 22% of students in the summer semester (5 out of 23) and 15% of students in winter semester (4 out of 27) addressed the ability to replay and pause videos on topics that were not well understood as very beneficial. Furthermore, the flexible access time was appreciated.

> *"Overall, everything was really very clear and extremely well done. In addition because you could watch the videos when you are awake and fit (and take breaks when your concentration drops), you absorb a lot more than usual. On the contrary, I had the feeling that I had just really understood many topics."*

While the weekly provision of the course units in the emergency remote [5] concept was perceived positively (for not being overwhelmed by materials), the daily release of the lecture units in the flipped classroom concept was criticized as the amount of material added up quickly leading to perceived time pressure. Furthermore, students criticized learning units with high information density. One student reported the "feeling of being overwhelmed". In this regard, students expressed their wish for more selection and clear focus on the essential content.

### 3.3.2. Usability and Structure

Most respondents agreed that the course was easy to use (summer 4.77 [0.69], winter 4.38 [0.88]/5 points, 1= worst, 5 = best). However, in the emergency remote [5] concept, several students reported having difficulties loading and playing the videos leading to increased processing time of the course. In some course units, poor picture and sound quality were also mentioned as disturbing factors. In addition, most respondents agreed with the statement that the course structure is clear (summer 4.64 [0.90], winter 4.52 [0.73]/5 points). 17% of students in summer (4 out of 23) and 15% of students in winter semester (4 out of 27) indicated a need for an additional scriptum of the lesson content providing more learning structure.

### 3.3.3. Clarity and Appropriateness

Clarity of the information provided was rated higher in the emergency remote [5] phase than in the flipped classroom concept (4.24 [1.04] vs. 3.60 [0.91]/5 points). Students had no problems answering the quiz after the course units in the emergency remote [5]

phase (1.74 [0.99]/5 points). In the free-text comments, the quizzes offered in the emergency remote [5] concept were perceived as enriching. According to the students, quizzes give the opportunity to test the learning success and give reward for learning and focusing on the videos.

> *"I often had the feeling that I concentrate much more on the videos, because everything could potentially come up in a question. And because it's a great feeling when you can answer the questions correctly right away!"*

While 86% of the students rated the amount of time as appropriate in the emergency remote [5] phase, this was the case for only 44% in the flipped classroom phase. A total of 77% and 22% rated the amount of material as adequate in the emergency remote [5] phase and the flipped classroom phase, respectively.

### 3.3.4. Missing of Face-to-Face Interaction

The statement "Direct interaction with lecturers in class was missing for better understanding" was answered heterogeneously (summer: 2.81 [1.44]; winter: 3.04 [1.04]/5 points). Accordingly, in the free-text comments, face-to-face teaching was not missed by the majority of the students.

A total of 13% of students in the summer semester (3 out of 23) and 7% of students in the winter semester (2 out of 27) missed a final examination. According to the students, the absence of an exam significantly restrains learning success. Consequently, they expressed their wish for an exam to ensure learning.

> *"I find the system without a final exam questionable. You only really prepare for the topics if you are asked about them in some way".*

### 3.3.5. Perceived Understanding

It was found that most of the students agreed with the statement "I have understood the course contents"; the students' agreement with this statement decreased in the winter semester (3.95 [1.24] vs. 3.16 [1.11]).

### 3.4. Project Phase ("Projects in Pathology")

The majority of students considered the group size to be appropriate, averaging 4.81 [0.49]/5 points, and the working atmosphere as pleasant, with 4.74 [0.71]/5 points.

In addition, in the project phase, the absence of face-to-face interaction was not considered to hinder cooperation (2.67 [1.33]/5 points), communication (2.63 [1.39]/5 points), or learning (2.63 [1.47]/5 points). However, classroom teaching was demanded by some students in the free-text comments.

Furthermore, the students were very satisfied with the support provided by the lecturers (4.89 [0.32]/5 points). A remarkable 100% of the students agreed—with a maximum number of 5 points—that the lecturers were available for questions, responded to suggestions, and formulated criticism fairly and constructively.

The knowledge gain with respect to research skills was rated with an average of 4.15 [0.91] points. In particular, knowledge gain was perceived highest in poster development (4.41 [0.57]) and the critical examination of scientific content (4.37 [0.74] points), followed by scientific research (4.26 [0.81]) and the management of scientific projects (3.82 [1.00]). The knowledge gain in terms of teamwork was rated slightly lower at 3.52 [0.94].

The students felt that their interest in pathology had increased via group work (4.04 [1.04]/5 points). Likewise, a fair portion of the students agreed that they possessed increased interest in scientific work (3.93 [1.14]/5 points).

The responses to the open question of what students liked best in the project phase revealed that students particularly appreciated the degree of interactivity. The opportunities to work together with fellow students on a project, to deal with a topic intensively, and to design a poster were perceived as very pleasant and instructive. A total of 30% of students (8 out of 27) mentioned that the focus on research (including guidance with respect to the

use of medical search engines, practicing critical reading, and designing posters)—which, according to the students, is often neglected in medical studies—was perceived as helpful.

> *"I found it very helpful to have a fixed course that encourages and accompanies research so that you are not completely on your own and lost."*

Motivation and support provided by the lecturers who "*encouraged students to remain curious*" was also highlighted by the students. The students found the schedule of the project phase to be reasonable. One student assessed the workload as appropriate, while another student found it to be inappropriately high.

As negative comments, four students (15%) expressed their wish for clearer communication, criticizing the goals and schedule for being unclear and claiming that the deadline for submitting the poster was communicated incorrectly. One student explained that the pandemic made it difficult to organize their responsibilities at times and expressed the wish for more structure.

The dismission of the poster presentation was also perceived negatively. Two students expressed their doubts as to the extent to which the mutual evaluation of the posters from other groups contributed to learning success. One student found the opportunity to evaluate posters of the other groups to be generally agreeable but described finding the evaluation difficult due to the specific character of the topics. Furthermore, the students found it difficult to discuss and evaluate the posters of the other groups, as many topics were quite specific. Therefore, the students proposed broadening the topics and assigning the same topic to several groups for elaboration.

## 4. Discussion

The requirement for emergency remote education to be set up quickly leads to several disadvantages with respect to course design and quality [5]. However, it is quite likely that the COVID-19 crisis will lead to an acceleration of the digitalization of medical education [45].

We analyzed the transition of an emergency remote education [5] pathology course to a novel flipped classroom approach, which included a general pathology/neuropathology e-learning module with meet-the-expert sessions, followed by a project phase integrating the intensified study of pathology using a problem-based learning approach [36] and research skills training; importantly, this approach can be switched from pandemic to non-pandemic mode at any time, as the project phase can also be held virtually.

### 4.1. Creating Asynchronous Content

In line with the literature, our results reflect the benefits of asynchronous teaching, (e.g., being able to study at one's own pace, without regard to the time and place) [46]. This flexibility is most important during the pandemic; therefore, asynchronous course offerings have been a common solution during the pandemic [46,47].

Interestingly, during the emergency remote education [5] phase, the students' satisfaction with the asynchronous "DigiPath module" was higher than it was with the conventional course offered previously. This changed in the flipped classroom concept when the satisfaction level with the asynchronous module was again about the same as the average satisfaction with the conventional course. One explanation for this could be that students were very grateful for the provision of a sufficient course during the crisis brought about by the first pandemic wave and, therefore, initially gave the course higher ratings. The drop to the baseline level could also be explained by the higher time pressure that the students faced. In addition, the elimination of the quiz, which was perceived very positively by the students, could lead to a decrease in the average score.

Although asynchronous teaching alone cannot replace existing medical teaching, we believe it is an important component of pandemic preparedness, as there is a shortage of personnel in particularly severe waves of the pandemic, which is firstly because personnel are needed in patient care and secondly because personnel themselves become infected with COVID-19 [48] and, therefore, cannot work. Furthermore, the students themselves may

fall ill and require patient care [49]. Therefore, it seems advisable to develop asynchronous content that can be used as flipped classroom content in non-pandemic times.

In line with the literature [50], we demonstrated that screencasts (which can be produced with high quality and with relatively little effort) are an appealing format for e-learning, especially when combined with other didactic methods. However, elaborated screencasts/videos with animations and drawings were considered to especially arouse interest. To enable the autonomy called for in self-determination theory [26] and in the theory of self-directed learning [24], teaching materials of sufficiently high quality must be provided. This was reflected in the students' enthusiasm for teaching units with screencasts and the commentary in which it was explained that simply uploading lecture slides does not lead to a sufficient increase in knowledge. In addition, the students showed satisfaction with being able to watch videos at their own pace, possibly as an indication for a sense of autonomy. Furthermore, in view of the agreement with the item "I have understood the course contents", we assume that the feeling of competence demanded in self-determination theory [26] could be achieved. However, the duration of the videos needs to be reflected, as length and, therefore, concentration is a limiting factor in e-learning [51]. Using simple lecture recordings may be helpful for students who are unable to attend a synchronous lecture or who require course content to be repeated. Therefore, we recommend providing screencasts and lecture recordings wherever possible.

The majority of students considered the course structure to be clear. As in any lesson, a logical structure and clearly defined objectives (which are always of particular importance in the development of a teaching unit [52]) are crucial factors that contribute to perceived learning success.

The quizzes between the units were appreciated and perceived as enriching as they were seen as an opportunity to test learning success and motivation to focus on the videos since "assessment drives learning" [53]. New course concepts may also lead to the need for new examination concepts; this should be considered early when designing new course concepts from emergency remote education [5] teaching content. In addition, sound and picture quality seem to affect the perception of the course and possibly the students' motivation as well. At least in our faculty, in the emergency phase, it was sometimes regarded as challenging to provide videos with high picture and sound quality.

Furthermore, the daily activation of the course units, which was perceived as overwhelming by the students, could have led to the above-mentioned decrease in satisfaction. It should also be ensured that the flipped classroom courses are manageable with respect to the time students spend away from the face-to-face courses and are not overloaded with such an excessive amount of content that a competitive situation emerges between the flipped classrooms and face-to-face courses. A similar concern about overload in flipped classrooms has already been expressed by Lafleur et al. [54]. Thus far, a reduction in the content in our course has not been possible for curricular reasons; thus, all content is still provided in an asynchronous format. Reducing online content will possibly become a relevant task in many medical schools when once again combining the newly created online content with more face-to-face teaching.

An important problem with respect to the creation of asynchronous course content can be lecturers' lack of availability in terms of time but also a lack of technical knowledge and technical equipment [55]. These problems have also appeared in our medical school. Therefore, we recommend providing training with respect to the use of media and online services regarding teaching and assistance in the use of digital technologies. It might be advisable to establish a working group supporting medical teachers in creating e-learning content as in our TUM Medical Education Center. However, this will not be possible at every medical school. Staff resources are also limited at our faculty; thus, we have now developed a wiki for teaching, which contains numerous instructions for creating online content. Such wikis have already proven useful in medical education [56] and might be a way for medical schools with limited staff resources to support their medical teachers in terms of online education. To increase efficiency, establishing individual rooms for digital teaching

equipped with a PC, camera, microphone, and green screen may constitute a cheaper alternative in order to avoid having to equip all lecturers with expensive technology [57].

Although the question regarding the lack of face-to-face interaction was answered unevenly by students, asynchronous teaching certainly cannot completely replace previous forms of medical teaching. This was recognized early on during the Emergency Remote Education phase. In their structured review, Bond et al. showed that "the most often mentioned type of [e-learning] tools were synchronous collaboration tools, especially video conferencing systems" [58]. Q&A sessions, whether face-to-face or virtual, are unavoidable in the long term and need to be considered in the planning process.

*4.2. Transition to Seminars/Project Phase*

Many faculties have developed new course concepts [58]. Our project phase was designed in a new manner, combining the intensified study of one pathology topic per group (using a problem-based learning approach [36]) on the one hand and teaching research skills on the other. In line with the related theory [36], the students appreciated the interactivity and the opportunity to work together with fellow students on a project during the project phase ("Projects in Pathology"). The fact that the students felt their interest in pathology had increased due to the group work might reflect their feeling of relatedness in accordance with self-determination theory [26]. Other faculties have already successfully implemented project-based learning units [59–62]. Ommering et al. [8] refer to Bandura's concept of self-efficacy, defined as an individual's belief in their own ability to accomplish a task, assuming that individuals are likely to engage in activities if they believe that they can succeed in these activities [63]. Ommering et al. assume that if students are more confident in their research abilities, the chance of their continued engagement in research increases [8]. In line with Ommering et al., our students stated that their interest in pathology had increased due to the group work along with their interest in scientific work. Therefore, integrating research skills into clinical courses seems to be a useful approach. Vereijken et al. already showed in their survey of undergraduate medical students that research practices foster students' research motivation and reinforce the notion that research is relevant to learning [25]. Furthermore, in the theory of connectivism [64], which is commonly used for online education, being able to see connections between fields, ideas, and concepts is regarded as a core skill [65]. However, students need to be trained in recognizing these connections to optimally study new content. Therefore, our goal was to link pathology content and research skills with case-based learning using a flipped classroom concept with seminars.

Other research skill courses have also proven to be successful [61,66]. Si et al. examined a course partly similar to "Projects in Pathology"; in their course, the study of research methods also took place prior to a small research project [61]. Research training sessions with flexible electives are also offered [67].

Working in small groups facilitates exchange between peers and the lecturer and enables closer mentoring. Thus, working in small groups increases the chances that students will gain more confidence regarding their research skills and fosters their positive self-efficacy beliefs [8]. When building new flipped classroom concepts out of emergency remote education [5] materials, we recommend keeping group sizes small and limiting them to eight people. The greatest challenge in this regard relates to personnel intensity. A possible way to reduce the personnel and cost pressure in this respect could be the peer teaching approach, which has already proven to be successful [68–70]. In order to facilitate the training of the lecturers and student assistants for such a course, it is worth creating an online course beforehand.

Additionally, not every medical school has a didactic center or a connection to a center for key competencies. However, for all the scientific content we integrated into the lessons, such as teamwork, the management of scientific projects, and scientific thinking, there are numerous sources of support in the literature [71–73] such that this content can also be taught

without the use of these backup facilities. Nevertheless, this will certainly be more time-consuming. However, this effort may be worthwhile in terms of pandemic preparedness.

*4.3. Transition and Future*

"DigiPath" and "Projects in Pathology" teaching have completely replaced the original course. The ability to switch between face-to-face and online mode in Q&A sessions and seminars is intended to allow for the quickest possible adaptation to possible further pandemic waves and any restrictions. The course will continue to be evaluated on an ongoing basis and optimized accordingly, as proposed by Kern et al. [34]. In our view, courses that allow for rapid switching between face-to-face and online teaching allow for superior pandemic preparedness. Nonetheless, we believe that further research is urgently needed in order to develop optimal strategies for pandemic preparedness in medical teaching.

*4.4. Strengths and Limitations*

Our study describes the transformation from a classical pathology course to a modern pathology course with a flipped-classroom approach, which can switch between fully digital and face-to-face modes at any time and gives practical and differentiated insights into the course development process. In addition, the present study shows possibilities and gives practical advice regarding how clinical teaching can be linked to the teaching of research skills.

However, there are weaknesses that need to be addressed. The students' response rate was quite low, which was probably because all the evaluations were completed simultaneously at the end of the semester and due to the length of this evaluation. In addition, different course designs (from the original course to the asynchronous emergency remote education [5] course (("DigiPath") to the flipped classroom concept (the combination of "DigiPath" and "Projects in Pathology")) were compared. On the one hand, these courses differ didactically; on the other hand, an additional component was added in the last course variant with respect to the training in scientific work. Although the courses are clearly structurally different, we consider a comparison of the students' experiences with the courses to be useful in order to show the different possible stages of such a transition and to demonstrate, on the basis of these examples, how future course concepts can be developed from emergency remote education [5] content. In addition, this is not a direct comparison of satisfaction with the courses, since the same students were not surveyed more than once, but a new group of students received the lessons each semester. We believe this is justified, since teaching the same content to the same students repeatedly would in itself have had an impact on satisfaction with the course. In addition, teaching a course multiple times to the same students during the pandemic does not seem practical.

We surmised that the students' agreement that they had understood the course content was an indication of a sense of competence, and we judged that their sense of having achieved an increased interest in scientific work through group work was an indication for the feeling of relatedness. We inferred that the comments about the being able to watch videos at one's own pace, as well as the preference for screencasts to facilitate autonomous learning, were indicators of a sense of autonomy. Nevertheless, the content of applied learning theories was not directly queried. Therefore, further research should focus more on the extent to which new course designs actually reflect learning theory constructs. Furthermore, we cannot provide examination results, as oral examinations had been suspended and the students only write the pathology exam after another integrated lecture in the following study year. Therefore, the impact on the students' knowledge cannot be proven.

**5. Conclusions**

On the one hand, the transition to Emergency Remote Education [5] materials can include asynchronous content that can be used in flipped classrooms. Screencasts, which

can be created with little technical effort, are suitable forms of presentation, along with scripts and case vignettes. Asynchronous content can make an important contribution to pandemic preparedness in further pandemic waves, as it enables flexible learning, which is at least partially independent of lecturers. Nevertheless, additional Q&A sessions are indispensable. A flipped classroom approach can help with respect to the use of learning materials in each phase of the pandemic. On the other hand, linking training in scientific work with content-related topics is a promising approach to increasing students' interest in scientific work and can be woven into new course formats. An important challenge for medical schools will be to train their lecturers as efficiently as possible to master the necessary technical skills.

**Supplementary Materials:** The following supporting information can be downloaded at: https://www.mdpi.com/article/10.3390/ejihpe13010008/s1, Supplemental Table S1: Overview of the online courses, Supplemental Table S2: example of lesson-planning outline (neuroendocrine tumors vs. neuroendocrine carcinomas) drafted during the planning of the project phase, Supplemental Table S3: Overall results of long survey, and Supplemental Table S4: Overall results of short survey (part of general semester evaluation).

**Author Contributions:** C.H.-L. designed the questionnaires, designed materials for the teaching of research skills, analyzed the results and wrote the manuscript, N.J. designed the questionnaires and wrote the manuscript, M.W.-M. gave substantial input to the course development and revised the manuscript critically, F.L.-S. was involved in the course design, the design of the study methods and revised the manuscript critically, K.S.-B. designed materials for the teaching of research skills and revised the manuscript critically, L.R. interpreted data and critically revised the manuscript, K.W. designed the questionnaires and wrote the manuscript, B.K. designed the new course concept and revised the manuscript critically, J.S. designed the new course concept, developed several new course content, supervised the study and revised the manuscript critically. All authors have read and agreed to the published version of the manuscript.

**Funding:** This research received no external funding.

**Institutional Review Board Statement:** The study was conducted in accordance with the Declaration of Helsinki, and approved by the Institutional Review Board (or Ethics Committee) of the Medical Faculty of the Technical University of Munich under the code 340/20 S-KH.

**Informed Consent Statement:** Informed consent was obtained from all subjects involved in the study.

**Data Availability Statement:** All relevant data can be found in the supplement in aggregated form, the publication of the original datasets is not possible due to German and faculty data protection requirements.

**Conflicts of Interest:** The authors declare no conflict of interest.

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
