# Peer review of "Design and Transition of an Emergency E-Learning Pathology Course for Medical Students—Evaluation of a Novel Course Concept"

_ejihpe, doi:10.3390/ejihpe13010008_

Round 1

Reviewer 1 Report

This article is devoted to one of the topical issues of modern education, namely, the organization of online learning, which is due to the emergence of the Covid-19 pandemic. A particularly problematic topic is the training of future doctors since people's lives and health depend on their competence. Therefore, this study has special relevance for education issues. The learning process is presented in great detail, and positive aspects of online learning are highlighted, for example, the ability to play a video, which contributes to a better consolidation of the material. The negative points of view of students on this form of education are also reflected.

Interesting from the point of view of the teacher is the description of the phases of the course implementation. training However, the authors did not explicitly express the question of ways of communication between the teacher and the student in addition to viewing the course. Were the cheats of the course students used in social networks, where students could discuss emerging issues with the teacher and other students? The question also arises whether a small number of students in groups respond to the usual education process or whether was it an emergency measure during the pandemic. The authors of the article did not assess how the results of the online learning period were compared to previous years. The authors presented decent results for students' education for the studied period. But I believe that they have been teaching for years and have comparative statistics. The authors should also note which elements of the course they have left online, and which elements of the course have now gone offline.

Author Response

We thank the reviewer for the thorough review of our manuscript and the very helpful feedback. We have done our utmost to respond to all comments to your complete satisfaction. However, if there are any further requests for changes, we will of course implement them.

Comment 1: This article is devoted to one of the topical issues of modern education, namely, the organization of online learning, which is due to the emergence of the Covid-19 pandemic. A particularly problematic topic is the training of future doctors since people's lives and health depend on their competence. Therefore, this study has special relevance for education issues. The learning process is presented in great detail, and positive aspects of online learning are highlighted, for example, the ability to play a video, which contributes to a better consolidation of the material. The negative points of view of students on this form of education are also reflected.

Response 1: We thank the reviewer for this encouraging assessment.

Comment 2: Interesting from the point of view of the teacher is the description of the phases of the course implementation. However, the authors did not explicitly express the question of ways of communication between the teacher and the student in addition to viewing the course. Were the cheats of the course students used in social networks, where students could discuss emerging issues with the teacher and other students?

Response 2: We thank the reviewer for this comment and have made additions in some passages to make this clearer:

“For these virtual question and answer meetings, Zoom ® (33) was used in the version licensed for our university and complying with local, German, and European data protection regulations. In addition, students could also email their questions to the respective lecturers. A question forum was also set up in the Moodle® (28) platform.” (p. 4, ll. 167-170)

“These trainings were organized together with and guided by TUM Center of Key Competencies, materials on teaching research skills were produced by TUM Medical Education Center.” NEW: “These trainings were held as face-to-face meetings as part of the regular pathology staff meetings. All training content was also made available to lecturers on the Moodle® (32) e-learning platform.” (p. 5, ll. 207-2011

Comment 3: The question also arises whether a small number of students in groups respond to the usual education process or whether was it an emergency measure during the pandemic.

Response 3: We thank the reviewer for this remark. The course has completely replaced the old offer, participation in the previous course offer is no longer possible since the implementation of the new concept. To make this clearer, we have added the following sentence: “It has replaced the previous original pathology course – the original course cannot be attended since then.” (p. 3, ll. 130-131)

Comment 4: The authors of the article did not assess how the results of the online learning period were compared to previous years. The authors presented decent results for students' education for the studied period. But I believe that they have been teaching for years and have comparative statistics.

Response 4: In our manuscript, we had used the evaluation from the winter semester 2019/2020 as a comparative grade, this was 1.94. This was a different evaluation process where each individual course unit was evaluated by the students, here we only have the overall student satisfaction as these surveys were much less detailed as they were collected for all courses throughout the program. After the reviewer's comment, we were very happy to go through the results of other semesters. We have now evaluated the overall results from the summer semester 2017 to the winter semester 2019/2020 that are still available to us and calculated an average value. This is 1.9 (best rating 1.85 (winter semester 2017/2018), worst average rating 1.98 in summer semester 2017). We have therefore added the following sentences: methods section: “This item had also been asked in previous semesters as part of the general evaluation. In order to achieve comparability with previous evaluation results, the average grade of all evaluations of the individual course units from the summer semester 2017 to the winter semester 2019/2020 (total semesters) was calculated.” (p. 7, ll. 269-273),   overall satisfaction: “The mean grade of all course evaluations of the original pathology course (from the summer semester 2017 to the winter semester 2019/2020) had been 1.9 (best average evaluation 1.85 in the winter semester 2017/2018, worst average evaluation 1.98 in the summer semester 2017). In the long survey, the asynchronous teaching units in the new emergency remote education (5) phase were rated better with an average grade of 1.51, followed by a 1.98 in the flipped classroom concept. The project phase was rated better with 1.44.“ (p.8, ll.308-313) We have changed figure 4 accordingly. We hope we could clarify this matter with these changes.

Comment 5: The authors should also note which elements of the course they have left online, and which elements of the course have now gone offline.

Response 5: We thank the reviewer for this important remark. Despite the risk of information overload, the content had to be maintained for curricular reasons. Therefore, in order not to confuse readers, we have changed the sentence: " The asynchronous contents were slightly modified, and the units were divided into five subjects: general pathology, inflammation/immunology, general tumorology, special tumorology and neuropathology. " (p.5, ll.192-194). Units were combined here, but content was not completely removed. We also picked this up in the discussion: " A reduction of the content in our course has not been possible so far for curricular reasons, all content will continue to be provided in the asynchronous part. Reducing online contents will possibly become a relevant task in many medical schools when combining the newly created online contents with more face-to-face teaching again." (p. 13, ll. 518-522).

Reviewer 2 Report

Reading the introduction part carefully, I found that the article is very well founded from a theoretical point of view, but still, I have a few question marks and a recommendation:

·        Line 80-82: we aimed to encourage medical students' interest in research as well as to increase their knowledge on research by engaging them in small research projects (8, 13, 25).  We aimed to encourage medical students' interest in research and increase their knowledge on research by engaging them in small research projects authors cited in the three papers, or was it a common aim?

·        Throughout the introduction part contains three different study objectives:  you aimed at a flipped classroom design that could be used under pandemic and non-pandemic circumstances; you aimed to meet the students' needs and strengthen their feelings of autonomy, competence and relatedness in order to enhance their intrinsic motivation and interest in research and you aimed to encourage medical students' interest in research as well as to increase their knowledge on research by engaging them in small research projects.  Do you not think it would be better to standardise the objectives of the study in a paragraph containing all of them?

1.      I understand that the purpose of the work was to investigate the students' satisfaction with the new course concept compared to the traditional teaching methods as well as the Emergency Remote Education course.  In addition, the students’ attitude towards the pathology and research after attending the new course should be evaluated.  Furthermore, it should be investigated if the students lack direct contact with the lecturers in the new course concept.  At the same time, we would like to describe the transformation of an originally classical pathology course via an Emergency Remote Education course to a new course model and thus, give suggestions on how to create new course concepts with Emergency Remote Education contents and which pitfalls should be considered.  A Novel Course Concept is covered in the article so please place the article in the specialised literature.

2.      Line 525: Furthermore, in the theory of connectivism (59).  Siemens G. Connectivism: A Learning Theory for the Digital Age http://www.  elearnspace.  org. Articles/connectivism htm (Accessed: 01/03/2007).  2005.  Please change the source access date.

3.      In several parts of the article, the authors discussed training and developing research skills (lines 25, 28, 33, 64, 66, 77, 78, 157, 185, 192, 206).  I understand that they used a blended learning concept, with case based learning (using patient centered cases) and developed a course curriculum integrating research skills and clinical knowledge.  The recommendation is to explain more clearly and in detail how this training and development of research skills was done.

4.      Please enter the sample size calculation in relation to the typology of the present study.

5.      Please specify more clearly the inclusion and exclusion criteria; respectively, from the total of 185 and 157-course participants who received both surveys (lines 246-249).

6.      The recommendation is that the conclusions should be shorter, to the point, in relation to the results obtained.

7.      The article was included to analyse the similarity coefficient in the Plagiarism CheckerX software, and please perform the following word reformulations that are in bold and italics:

·        therapeutic decisions (10). Also, the Canadian Medical Education Directives for Specialists (CanMEDS) emphasises the importance of research by defining it as one of the core competencies (11).

Author Response

We thank the reviewer for the thorough and extensive review of our manuscript. We have done our best to answer all comments to your full satisfaction. However, if there are any further requests for changes, we will of course implement them.

Comment 1: Reading the introduction part carefully, I found that the article is very well founded from a theoretical point of view, but still, I have a few question marks and a recommendation:

Response 1:

We thank the Reviewer for this encouraging feedback.

Comment 2: Line 80-82: we aimed to encourage medical students' interest in research as well as to increase their knowledge on research by engaging them in small research projects (8, 13, 25).  We aimed to encourage medical students' interest in research and increase their knowledge on research by engaging them in small research projects authors cited in the three papers, or was it a common aim?

Response 2: We thank the reviewer for pointing out that an explanation would be helpful here. Sources 8 and 25 deal with course concepts that can be used to teach research skills. Source 13 deals more with the necessity of teaching such skills, which we also saw expressed in the sentence, which is why we have also used the reference here. However, since references 8 and 25 sufficiently support the sentence, we have removed reference 13 after this sentence and changed the sentence as follows: “As proposed by Ommering et al., and following the findings of Vereijken et al., we thus aimed to encourage medical students’ interest in research as well as to increase their knowledge on research by engaging them in small research projects (8, 25).” (p. 2, ll. 80-83)

Comment 3: Throughout the introduction part contains three different study objectives:  you aimed at a flipped classroom design that could be used under pandemic and non-pandemic circumstances; you aimed to meet the students' needs and strengthen their feelings of autonomy, competence and relatedness in order to enhance their intrinsic motivation and interest in research and you aimed to encourage medical students' interest in research as well as to increase their knowledge on research by engaging them in small research projects.  Do you not think it would be better to standardise the objectives of the study in a paragraph containing all of them?

Response 3: We thank the reviewer for this advice, we are also very interested in a good clarity. We believe that a longer introduction to the objectives is necessary, which is why we did not want to shorten the introduction. However, in order to make the objectives clearer when creating the course, we are now very happy to include a summary of these objectives in the introduction: “Taken together, the new course design was intended to serve three goals: First, a   flipped classroom design for the pathology course that could be used under pandemic and non-pandemic circumstances for long-term use was to be produced. Second, medical students’ interest in research and their knowledge on research were to be improved by engaging them in small research projects. Third, in doing so, the students’ feelings of autonomy, competence and relatedness were to be strengthened in order to enhance their intrinsic motivation and interest in research.” (p.3, ll. 103-109)

Comment 4:      I understand that the purpose of the work was to investigate the students' satisfaction with the new course concept compared to the traditional teaching methods as well as the Emergency Remote Education course.  In addition, the students’ attitude towards the pathology and research after attending the new course should be evaluated.  Furthermore, it should be investigated if the students lack direct contact with the lecturers in the new course concept.  At the same time, we would like to describe the transformation of an originally classical pathology course via an Emergency Remote Education course to a new course model and thus, give suggestions on how to create new course concepts with Emergency Remote Education contents and which pitfalls should be considered.  A Novel Course Concept is covered in the article so please place the article in the specialised literature.

Response 4: We thank the reviewer for this comment and agree that our article can be further contextualized here within existing literature on online pathology courses. We believe that elaboration on the types of individual courses would go too far and reduce the fluid readability of the text. Nevertheless, a reference to selected literature here seems very useful. We have therefore supplemented the sentences as follows and inserted four appropriate references: “During the COVID-19 pandemic, many new course concepts e-learning course concepts have emerged, including pathology (28-31). Based on the experience we have gained, we would like to give suggestions on how to create new course concepts out of Emergency Remote Education contents and which pitfalls should be considered.” (p. 3, ll. 117-120)

Comment 5:      Line 525: Furthermore, in the theory of connectivism (59).  Siemens G. Connectivism: A Learning Theory for the Digital Age http://www.  elearnspace.  org. Articles/connectivism htm (Accessed: 01/03/2007).  2005.  Please change the source access date.

Response 5: We thank the reviewer for pointing this out. As this weblink is not valid any more, we have replaced it by a new link to the source (https://www.itdl.org/Journal/Jan_05/article01.htm )with the new access date. (p. 19, ll. 798)

Comment 6: In several parts of the article, the authors discussed training and developing research skills (lines 25, 28, 33, 64, 66, 77, 78, 157, 185, 192, 206).  I understand that they used a blended learning concept, with case based learning (using patient centered cases) and developed a course curriculum integrating research skills and clinical knowledge.  The recommendation is to explain more clearly and in detail how this training and development of research skills was done.

Response 6:  We thank the reviewer for this note. We are also interested in a good comprehensibility of the contents. We had tried to explain this in section 2.1.2.2. We have gladly added the following sentences and an additional reference to Supplemental Table S2, which shows an example workflow. “For these skills, the trainers gave explanations and short introductions to the students. Subsequently, the students had to implement this by means of homework, presentations, group work and the final poster creation, whereby the progress was controlled by the trainers during the seminar sessions. Supplemental Table S2.“ (p. 6, ll. 225-229)

Comment 7:   Please enter the sample size calculation in relation to the typology of the present study.

Response 7: We thank the reviewer for this remark. We agree that sample size calculation should also be performed in educational research, which is rarely done in practice. Due to the fact that the student groups were predefined for curricular reasons (numbers could not be adjusted) and no case-control design could be applied because the course evolved from semester to semester and all students enrolled in the course were required to participate in the respective course, sample size calculation was not feasible in this case. However, we agree that this should be mentioned and added the following sentence to the statistics paragraph: “Because the student cohorts were predefined by the curriculum and could not be changed, a sample size calculation was not performed for this observational study.” (p 8, ll. 297-299)

Comment 8: Please specify more clearly the inclusion and exclusion criteria; respectively, from the total of 185 and 157-course participants who received both surveys (lines 246-249).

Response 8: We thank the reviewer for this advice. All students in the course were invited to participate in the survey in both semesters. Exclusions were not necessary because all participants had given their written informed consent and no students were under 18 years of age in these semesters. We have tried to clarify this as follows: “In summer semester 2020 both surveys were sent to all 185 course participants, in winter semester 2020/2021, to all 157 participants. (p. 7, ll. 276-277); Since all participants had given their written informed consent, no exclusions were necessary. If individual questions were not answered, these were indicated as missing in the analysis. (pp. 7-8, ll. 281-283)

Comment 9: The recommendation is that the conclusions should be shorter, to the point, in relation to the results obtained.

Response 9: We thank the reviewer for this important comment. We have tried to shorten the discussion and strengthened the textual links to the results at certain points. However, since the other two reviewers did not have any objections here, we could not completely change the discussion and have therefore tried to find a balance between the views of all reviewers with the changes now made. (pp. 12-16, ll. 447-642)

Comment 10: The article was included to analyse the similarity coefficient in the Plagiarism CheckerX software, and please perform the following word reformulations that are in bold and italics:

  • therapeutic decisions (10). Also, the Canadian Medical Education Directives for Specialists (CanMEDS) emphasises the importance of research by defining it as one of the core competencies (11).

Response 10: We thank the reviewer for the remark. We assume that the similarity is from our earlier preprint. However, to prevent misunderstandings here, we were very happy to change the sentence: “This is underlined by the Canadian Medical Education Directives for Specialists (CanMEDS) highlighting the importance of research as one of the core competencies (11).” (p.2, ll.71-73)

Reviewer 3 Report

Great work and rigorous methods. Consider removing or cleaning up Figure 1A (writing is hard to read). Figure 1B, please edit to translate the captions in English. Optionally, consider including PMID 2366598, one of the earlier papers about a structured digital-based online pathology curriculum arising within the context of Covid-19. 

Author Response

We thank the reviewer for the thorough review of our manuscript and the very encouraging assessment. We were very happy to make the requested changes. Should the reviewer nevertheless wish to see further changes, we will of course be very happy to implement them.

Comment 1: Great work and rigorous methods.

Response 1: We thank the reviewer for this positive and encouraging assessment.

Comment 2: Consider removing or cleaning up Figure 1A (writing is hard to read). Figure 1B, please edit to translate the captions in English.

Response 2: We thank the reviewer for this valuable assessment and share the view that the sample figure could not be interpreted well as it was. We have now changed Figure 1A and 1B and hope to provide a better illustration of the video examples.

Comment 3: Optionally, consider including PMID 2366598, one of the earlier papers about a structured digital-based online pathology curriculum arising within the context of Covid-19. 

Response 3: We thank the reviewer for this very good additional reference and were very happy to cite it as follows: “During the COVID-19 pandemic, many new course concepts e-learning course concepts have emerged, including pathology (28-31). Based on the experience we have gained, we would like to give suggestions on how to create new course concepts out of Emergency Remote Education contents and which pitfalls should be considered.” (p. 3, ll. 117-120) As reviewer 2 had also asked to set the novel course concept more in the context of the current literature we added three other references as well.

Round 2

Reviewer 2 Report

I congratulate the authors for their hard work producing this article with many novel elements.